# Biosynthesis of helvolic acid and identification of an unusual C-4-demethylation process distinct from sterol biosynthesis

Jian-Ming Lv[1], Dan Hu [1], Hao Gao [1], Tetsuo Kushiro[2], Takayoshi Awakawa[2], Guo-Dong Chen[1], Chuan-Xi Wang [1], Ikuro Abe [2] & Xin-Sheng Yao[1]

Fusidane-type antibiotics represented by helvolic acid, fusidic acid and cephalosporin $P_1$ are a class of bacteriostatic agents, which have drawn renewed attention because they have no cross-resistance to commonly used antibiotics. However, their biosynthesis is poorly understood. Here, we perform a stepwise introduction of the nine genes from the proposed gene cluster for helvolic acid into *Aspergillus oryzae* NSAR1, which enables us to isolate helvolic acid (~20 mg $L^{-1}$) and its 21 derivatives. Anti-*Staphylococcus aureus* assay reveals that the antibacterial activity of three intermediates is even stronger than that of helvolic acid. Notably, we observe an unusual C-4 demethylation process mediated by a promiscuous short-chain dehydrogenase/reductase (HelC) and a cytochrome P450 enzyme (HelB1), which is distinct from the common sterol biosynthesis. These studies have set the stage for using biosynthetic approaches to expand chemical diversity of fusidane-type antibiotics.

[1] Institute of Traditional Chinese Medicine and Natural Products, College of Pharmacy/Guangdong Province Key Laboratory of Pharmacodynamic Constituents of TCM and New Drugs Research, Jinan University, Guangzhou 510632, People's Republic of China. [2] Graduate School of Pharmaceutical Sciences, The University of Tokyo, 7-3-1 Hongo, Bunkyo-ku, Tokyo 113-0033, Japan. Jian-Ming Lv and Dan Hu contributed equally to this work. Correspondence and requests for materials should be addressed to H.G. (email: tghao@jnu.edu.cn) or to I.A. (email: abei@mol.f.u-tokyo.ac.jp) or to X.-S.Y. (email: tyaoxs@jnu.edu.cn)

Triterpenoids possess a variety of pharmacological activities and serve as an invaluable source of pharmaceuticals[1]. Ginsenosides and ganoderic acids are drug candidates[2, 3], and ginsenoside Rg3 has been clinically approved to treat cancers in China. However, extremely low amounts of these medicinal constituents in natural hosts have significantly hampered their applications[4, 5]. Combinatorial biosynthesis is a feasible option to improve the yields of target compounds and expand chemical diversity, which however requires a thorough understanding of their biosynthesis. To date, only a few triterpenoids such as ginsenosides[6] and betulinic acid[7] have been demonstrated.

Fusidane-type antibiotics, a group of fungi-derived triterpenes, originate from protosta-17(20)$Z$,24-dien-3$\beta$-ol (**2**) with unique tetracyclic skeleton, which is formed by enzymatic cyclization of (3$S$)-2,3-oxidosqualene[8]. Thus, after the formation of the tetracyclic C-20 protosteryl tertiary intermediate cation[9], the cyclization reaction is terminated by elimination of H-17 to form a double-bond between C-17 and C-20 without the backbone rearrangement. Helvolic acid (**1**)[10], fusidic acid[11], and cephalosporin P$_1$[12] are the most representative fusidane-type antibiotics (Fig. 1a), all of which exhibit potent activity against Gram-positive bacteria[13]. Amongst them, fusidic acid has been used to treat staphylococcal skin infections in Europe for nearly 60 years[14]. In the United States, the recent phase II clinical trial has demonstrated that fusidic acid showed comparable efficacy to linezolid for treatment of acute bacterial skin and skin structure infections (ABSSSIs)[15]. Along with the advancement of global antimicrobial resistance, fusidane-type antibiotics have drawn renewed attentions because they have no cross-resistance to commonly used antibiotics[14, 16].

The remarkable bioactivity and little cross-resistance of fusidane-type antibiotics have inspired many attempts to search for more active congeners. Great efforts have been focused on modifications of fusidane skeleton by means of chemical derivatization and biotransformation[13, 17]. Exploring fusidane-type antibiotics biosynthesis could offer an opportunity to yield structurally novel derivatives, but the genetic and molecular basis for the biosynthesis of fusidane-type antibiotics remained obscure until 2009 when helvolic acid putative biosynthetic gene cluster consisting of nine genes (Fig. 1b and Table 1) was identified from the genome sequence of *Aspergillus fumigatus* Af293[18, 19]. Preliminary heterologous expression studies in *Saccharomyces cerevisiae* revealed that HelA (oxidosqualene cyclase, OSC), functioned to convert (3$S$)-2,3-oxidosqualene into protosta-17(20)$Z$,24-dien-3$\beta$-ol (**2**), which underwent dehydrogenation to form 3-keto (**3**) by HelC (short-chain dehydrogenase/reductase,

SDR), or oxidation to generate 4$\beta$-carboxylic acid (**4**) by HelB1 (cytochrome P450, P450)[18]. Nevertheless, it is still not clear whether the proposed nine-gene cluster is enough for helvolic acid biosynthesis, and if so, what the functions and reaction order of the six other genes are, and whether biosynthetic intermediates with more potent antibacterial activity than the end product helvolic acid exist.

In this work, we first introduce all the nine genes into the heterologous expression system *A. oryzae* NSAR1[20], and observe the production of helvolic acid (~20 mg L$^{-1}$), confirming that the putative gene cluster is sufficient for helvolic acid biosynthesis. Then, we determine the function of each gene and reaction sequences by reconstitution of the nine genes one by one in *A. oryzae* NSAR1, which enables us to isolate 21 derivatives and find three intermediates with more potent activity than helvolic acid. Remarkably, we observe an unusual C-4 demethylation process that is distinct from the sterol biosynthesis. These findings have provided a basis for constructing fusidane-type derivatives using biosynthetic approaches.

## Results

**Heterologous expression of the 9-gene cluster in *A. oryzae*.** For convenience, we designated the reported helvolic acid gene cluster from *A. fumigatus* Af293 as the *hel* cluster and renamed all nine genes (Fig. 1b and Table 1). In order to determine whether the proposed gene cluster is sufficient to produce helvolic acid, we constructed an *A. oryzae* NSAR1 transformant strain AO21 co-expressing all the nine genes. After AO21 cultured in the induction medium for 6 days, the mycelia extract was analyzed by high performance liquid chromatography (HPLC). As a result, an additional peak (**1**) (~20 mg L$^{-1}$) was detected (Fig. 2h). According to high resolution electrospray ionization mass spectroscopy (HRESIMS) and nuclear magnetic resonance (NMR), **1** was established as helvolic acid. Therefore, we demonstrated that the presumed *hel* cluster was sufficient for the biosynthesis of helvolic acid, setting the stage for uncovering the detailed biosynthetic assembly line for helvolic acid.

**Stepwise reconstitution of helvolic acid biosynthesis.** Previously, the functions of HelA (OSC), HelB1 (P450), and HelC (SDR) have been studied through reconstitution in *S. cerevisiae*[18]. Consistently, we also observed the production of protosta-17(20)$Z$,24-dien-3$\beta$-ol (**2**) and its 3-keto derivative **3** when co-expressing *helA* and *helC* in *A. oryzae* NSAR1 (Supplementary

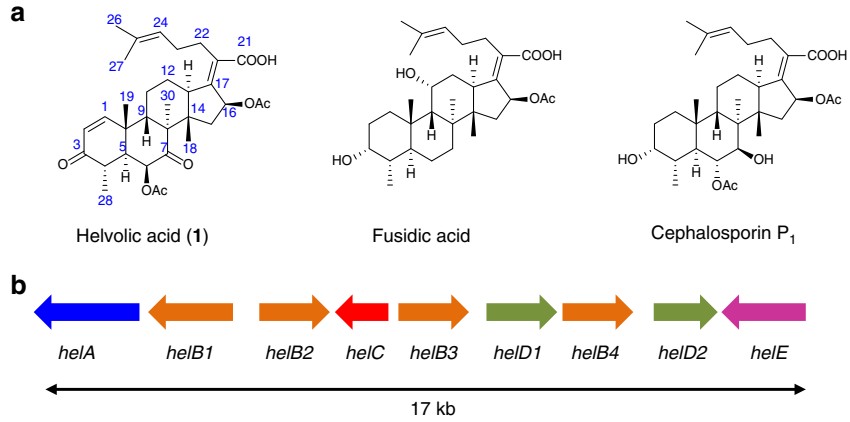

**Fig. 1** Representative fusidane-type antibiotics and biosynthetic gene cluster of helvolic acid (**1**). **a** Structures of helvolic acid (**1**), fusidic acid and cephalosporin P$_1$; **b** Gene map of the helvolic acid gene cluster from *A. fumigatus* Af293

**Table 1 Putative functions of genes in the *hel* cluster**

| Gene | Genbank accession number | Putative function |
|---|---|---|
| *helA* | XM_746263 | Oxidosqualene cyclase |
| *helB1* | XM_746262 | Cytochrome P450 |
| *helB2* | XM_746261 | Cytochrome P450 |
| *helC* | XM_746260 | Short-chain dehydrogenase/reductase |
| *helB3* | XM_746259 | Cytochrome P450 |
| *helD1* | XM_746258 | Acyltransferase |
| *helB4* | XM_746257 | Cytochrome P450 |
| *helD2* | XM_746256 | Acyltransferase |
| *helE* | XM_746255 | 3-Ketosteroid-$\Delta^1$-dehydrogenase |

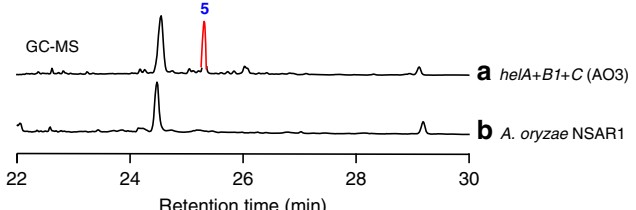

**Fig. 3** GC–MS analysis of mycelia extract. **a** *A. oryzae* harboring *helA*, *helB1* and *helC*; **b** *A. oryzae*

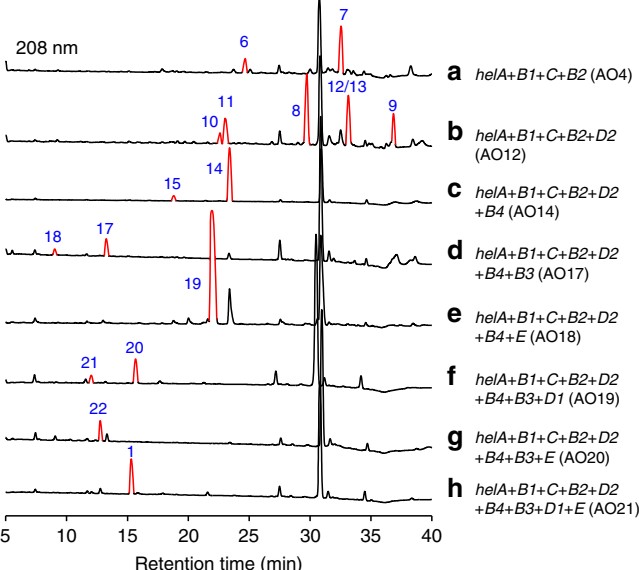

**Fig. 2** HPLC analysis of mycelia extract from different transformants. **a** *A. oryzae* harboring *helA*, *helB1*, *helC* and *helB2*; **b** *A. oryzae* harboring *helA*, *helB1*, *helC*, *helB2* and *helD2*; **c** *A. oryzae* harboring *helA*, *helB1*, *helC*, *helB2*, *helD2* and *helB4*; **d** *A. oryzae* harboring *helA*, *helB1*, *helC*, *helB2*, *helD2*, *helB4* and *helB3*; **e** *A. oryzae* harboring *helA*, *helB1*, *helC*, *helB2*, *helD2*, *helB4* and *helE*; **f** *A. oryzae* harboring *helA*, *helB1*, *helC*, *helB2*, *helD2*, *helB4*, *helB3* and *helD1*; **g** *A. oryzae* harboring *helA*, *helB1*, *helC*, *helB2*, *helD2*, *helB4*, *helB3* and *helE*; **h** *A. oryzae* harboring *helA*, *helB1*, *helC*, *helB2*, *helD2*, *helB4*, *helB3*, *helD1* and *helE*

Fig. 1c), and **4** upon co-expression of *helA* and *helB1* (Supplementary Fig. 1b). Considering that HelB1 and HelC are responsible for the generation of 4β-carboxylic acid and 3-keto, respectively, it is conceivable that HelB1 and HelC could function together to give a β-keto carboxylic acid intermediate, which would undergo decarboxylation to remove the C-4β methyl group. In order to verify the hypothesis, we simultaneously introduced *helA*, *helB1*, and *helC* into *A. oryzae* NSAR1 to generate a transformant strain AO3. Upon gas chromatograph–mass spectrometer (GC–MS) analysis, an additional peak at *m/z* 410 was observed in the mycelia extract of AO3, but not in the parent strain *A. oryzae* NSAR1 (Fig. 3 and Supplementary Fig. 2), which was also confirmed by HPLC analysis of AO3 (Supplementary Fig. 1a). HRESIMS and NMR analyses revealed that the additional peak was the demethylated product **5** (Fig. 4). These results clearly indicated that the HelB1 and HelC together could trigger the 4β-demethylation via oxidative decarboxylation (Fig. 5a),

which is distinct from the sterol biosynthesis by 4α-demethylation (Fig. 5b)[21, 22].

Assuming that **5** was the biosynthetic intermediate of helvolic acid, among the remaining six genes, only the P450 enzymes HelB2-4 and the dehydrogenase HelE might be engaged in the next modification. Thus we individually introduced *helB2*, *helB3*, *helB4*, and *helE* into the strain AO3 to generate transformants AO4–AO7. HPLC analyses of these strains showed that only introduction of *helB2* (AO4) gave two additional peaks **6** and **7** (Fig. 2a and Supplementary Fig. 3). After isolation, both **6** and **7** were determined to bear 16β-hydroxyl group by NMR analysis, demonstrating that HelB2 (P450) is responsible for hydroxylation of the C-16β position (Fig. 4). **6** ought to be directly derived from **4** under the action of HelB2, while **7** could be generated by two plausible routes, i.e., hydroxylation of **5** by HelB2 or decarboxylation of **6** by HelC.

Similarly, supposing that **6** or **7** was the biosynthetic intermediate of helvolic acid, all the remaining five genes could be involved in the next reaction. We hence individually transformed each of the five genes into the strain AO4 to generate transformants AO8–AO12. We observed that only introduction of *helD2* (AO12) led to the appearance of additional peaks in the HPLC profile (Fig. 2b and Supplementary Fig. 4). NMR structural characterization showed that **8** and **9** were the corresponding 16-*O*-acetylated products of **6** and **7**, indicating that HelD2 (acyltransferase, AT) accounts for the acetylation of C-16β hydroxyl group (Fig. 4). For the generation of **9**, two possible pathways may also exist as in the case of **7**. Along with **8** and **9**, four other products **10**, **11**, **12**, and **13** were also isolated in the strain AO12, which appeared as two pairs of epimers (Fig. 2b and Supplementary Fig. 4). These compounds were possibly derived from **8** and **9** by acetylation-elimination followed by the rearrangement of the double bond between C-17 and C-20 to C-16 and C-17, and quenching of the C-20 cation with a water molecule (Fig. 5c, d). A similar mechanism has been employed in the biosynthesis of anditomin[23] and pyrroindomycins[24].

Owing to lack of hydroxyl group in the structures of **8** and **9**, we next constructed the six gene-harboring transformants AO13–AO15 by means of individual introduction of *helB3*, *helB4*, and *helE* into the strain AO12. Only the strain AO14 containing *helB4* afforded additional products **14** and **15** (Fig. 2c and Supplementary Fig. 5). **14** was featured with the carboxyl group at the C-20 position, indicating that HelB4 (P450) was responsible for converting C-20 methyl group to carboxylic acid (Fig. 4). And the minor product **15** was elucidated as the 3-keto reduced form of **14**. Unexpectedly, we did not detect the HelB4-oxidized product of **8** (compound **16** in Fig. 4) in AO14. To verify the possibility that the oxidized product **16** was quickly converted to **14** by HelC, we established a *helC*-lacking transformant AO16 harboring *helA*, *helB1*, *helB2*, *helD2*, and *helB4*, and observed the accumulation of **16** in its mycelia extract (Supplementary Fig. 6). These results indicated that HelB4 could oxidize **8** into **16**, which was then rapidly converted to **14** by HelC.

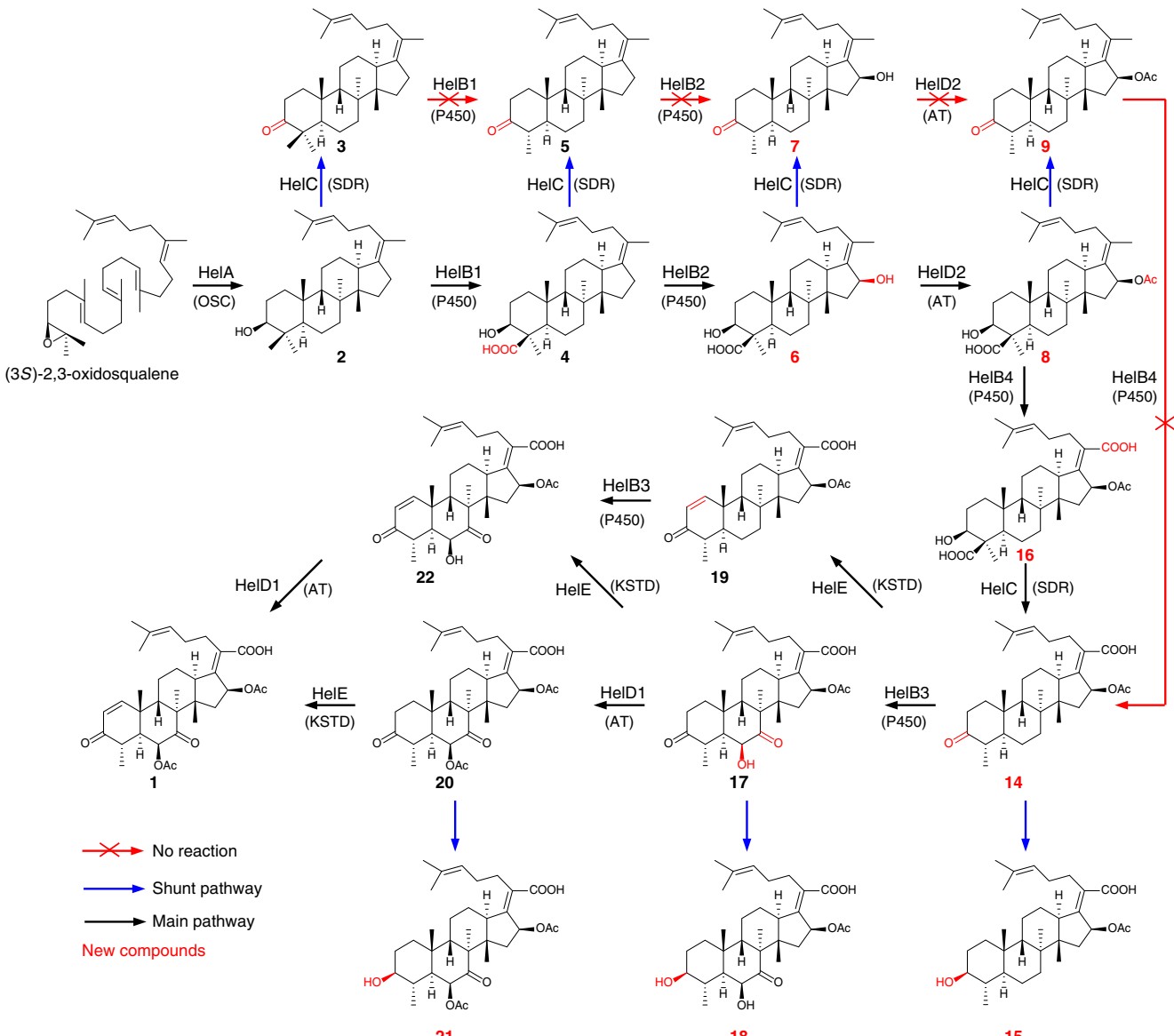

**Fig. 4** Complete biosynthetic pathway of helvolic acid. Biosynthesis of helvolic acid includes nine steps from the universal triterpenoid progenitor (3 S)-2,3-oxidosqualene. Following the cyclization to the tetracyclic protosta-17(20)Z,24-dien-3β-ol (**2**) by HelA, HelB1-mediated and HelB2-mediated oxidation at C-4 and C-16, HelD2-dependent acetylation of 16-OH, oxidation of C-21 by HelB4, and HelC-dependent oxidative decarboxylation yield the fusidane skeleton **14**, which is further modified in three additional steps mediated by HelB3, HelD1, and HelE to give helvolic acid. The premature decarboxylation (or dehydrogenation) by HelC prevents HelB1-mediated, HelB2-mediated, HelD2-mediated, and HelB4-mediated tailoring

Among the remaining three enzymes, only HelB3 (P450) and HelE (3-ketosteroid-Δ¹-dehydrogenase, KSTD) might function after HelB4. We then introduced *helB3* and *helE* into the strain AO14, respectively. The *helB3* containing transformant strain AO17 yielded **17** (Fig. 2d and Supplementary Fig. 7), which possesses the C-6 hydroxyl and C-7 carbonyl groups, thus revealing that HelB3 is responsible for the dual oxidation of C-6 and C-7 (Fig. 4). Similarly, we also isolated the 3-keto reduced product of **17** (compound **18**) (Fig. 2d and Supplementary Fig. 7). In addition to *helB3*, introduction of *helE* (AO18) into AO14 also resulted in the production of **19** (Fig. 2e and Supplementary Fig. 7), indicating that HelE catalyzes the dehydrogenation between C-1 and C-2 (Fig. 4). As in the case of AO14, we also did not observe the HelB3 or HelE-tailored products of **16**. To verify this, we also established the *helC*-lacking strain AO22 harboring *helA*, *helB1*, *helB2*, *helD2*, *helB4*, *helB3*, and *helE*. Compared with AO16, AO22 did not produce any additional products in both

medium and mycelia (Supplementary Fig. 8). It is unequivocal that unlike the enzymes HelB1, HelB2, HelD2, and HelB4 in the early stages of helvolic acid biosynthesis, HelB3 and HelE could only accept the decarboxylated product **14** as a substrate.

Subsequently, we expressed *helD1* or *helE* in AO17 to generate transformants AO19 and AO20. The mycelia extract of the transformant strain AO19 was analyzed, and two additional peaks emerged in the HPLC profile (Fig. 2f and Supplementary Fig. 9). Evaluation of HRESIMS and NMR led to the structural assignment of **20** as the C-6 acetylated product of **17**, confirming that HelD1 (AT) acts on the C-6 hydroxyl (Fig. 4). Additionally, **21** was determined to be the reduced product of **20** (Fig. 2f and Supplementary Fig. 9). On the other hand, the transformant strain AO20 harboring *helE* gave **22** (Fig. 2g and Supplementary Fig. 9), which might be also derived from oxidation of **19** by HelB3 (Fig. 4). Finally, the transformant strain AO21 harboring all the nine genes could produce the final product helvolic acid

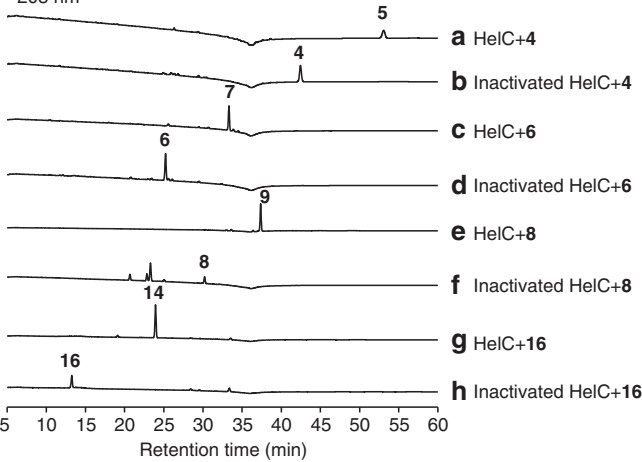

**Fig. 5** Proposed reaction mechanisms. **a** C-4 demethylation process in helvolic acid biosynthesis; **b** C-4 demethylation process in sterol biosynthesis; **c** Proposed mechanism for transformation of **8** to **10**/**11**; **d** Proposed mechanism for transformation of **9** to **12**/**13**

(~20 mg L$^{-1}$) (Fig. 2h and Supplementary Fig. 9). Taken together, we have established the complete biosynthetic pathway for helvolic acid and obtained as many intermediates and shunt products as possible (Fig. 4).

**An unusual C-4 demethylation process by HelB1 and HelC**. The heterologous expression in *A. oryzae* NSAR1 revealed that HelB1 and HelC were responsible for the removal of the C-4β methyl group, which is distinct from the sterol C-4 demethylation process (Fig. 5a, b). To elucidate which occurs first: oxidation of C-4β methyl group by HelB1 or HelC-mediated dehydrogenation, His$_6$-tagged HelC was expressed in *Escherichia coli* BL21(DE3) and purified (Supplementary Fig. 26). When we incubated HelC with substrate **4** in the presence of cofactor NAD$^+$ for 12 h, the decarboxylated product **5** was detected, but not in the negative control containing inactive HelC (Fig. 6a, b). However, when we incubated HelB1-expressing strain AO23 with substrate **3**, no product corresponding to **5** was detected based on liquid chromatograph–mass spectrometer (LC–MS) analysis (Supplementary Figs. 10 and 11). To rule out the possibility caused by the inactive expression of HelB1, we also fed this strain with substrate **2**, and we did find its oxidized product **4** (Supplementary Figs. 12 and 13). These results unambiguously indicated that HelC-mediated dehydrogenation follows HelB1-mediated oxidation of C-4β methyl, which is similar to the reaction sequence in the sterol biosynthesis[21, 22].

In the sterol biosynthesis, the loss of two C-4 methyl groups is achieved by the successive action of C-4 sterol methyloxidase (SMO), C-3 sterol dehydrogenase (3βHSD), and 3-keto reductase (3SR)[21, 22], while in the biosynthesis of helvolic acid, it is possible that the C-4 demethylation may take place at different stages as a result of the broad substrate range of HelC (Fig. 4). To verify these, we incubated recombinant HelC with substrates **6**, **8**, or **16**, and we found that HelC could convert **6**, **8** and **16** to the

**Fig. 6** HPLC analysis of in vitro enzymatic reactions. **a 4** with HelC; **b 4** with inactivated HelC; **c 6** with HelC; **d 6** with inactivated HelC; **e 8** with HelC; **f 8** with inactivated HelC; **g 16** with HelC; **h 16** with inactivated HelC

corresponding decarboxylated products **7**, **9** and **14**, respectively (Fig. 6c–h). These results demonstrated that HelC could work on multiple substrates to trigger the oxidative decarboxylation at different stages during the biosynthesis of helvolic acid.

The production of **7**, **9**, and **14** by HelC raised a doubt that the biosynthetic routes including HelB2-mediated conversion of **5** to **7**, HelD2-mediated conversion of **7** to **9**, and HelB4-mediated conversion of **9** to **14** may not exist. To test this hypothesis, feeding experiments for HelB2, HelB4 and HelD2 were carried out. The active expression of HelB2 in AO24, HelB4 in AO25, and HelD2 in AO26 were confirmed by feeding them with substrates **4**, **8**, and **6**, respectively (Supplementary Figs. 14, 15,

18, 19, 22, and 23). However, the transformant strains AO24 and AO25 could not generate 7 or 14 when supplemented with 5 and 9, respectively (Supplementary Figs. 16, 17, 20, and 21). These results suggested that the decarboxylation would prevent the substrates from modification by HelB2 and HelB4. Because 9 is hardly ionized, we could not confirm whether a very small amount of 7 was converted to 9 by AO26 using LC–MS analysis, but we did not observe any peak corresponding to 9 in the HPLC profile of AO26 fed with 7 (Supplementary Figs. 24 and 25), which indicates that HelD2 is likely to only accept the carboxylic derivative 6 (Fig. 4).

**Three intermediates are more active than helvolic acid**. In the course of our work, we obtained helvolic acid (~20 mg L$^{-1}$) and its 21 derivatives via reconstitution in *A. oryzae*, which provided an opportunity for structure-activity relationship analysis. We tested their inhibitory effects against *Staphylococcus aureus* 209P by the 2-fold dilution method (Table 2). Most compounds showed prominent inhibitory activities against this bacterium. Among them, 17, 19, and 22 showed even stronger activity than helvolic acid, and compound 22 exhibited the strongest antibacterial activity with minimum inhibition concentration (MIC) of 0.5 μg mL$^{-1}$. The structure-activity relationship analysis revealed that both C-20 carboxyl group and 3-keto were important for the antibacterial activity, which is consistent with previous findings[13]. In addition, we firstly found that acetylation of the C-6 hydroxyl group was detrimental to the activity, as 17 and 22 exhibited higher activity than their corresponding acetylated products 20 and helvolic acid, respectively. These facts suggested that the end products are not always to be the most active molecules, and using biosynthetic approaches to access intermediates would be an effective way to obtain bioactive analogs.

## Discussion

In this work, we have systematically unraveled the complete biosynthetic pathway for fusidane-type antibiotic helvolic acid by in vivo and in vitro experiments. Biosynthesis of helvolic acid includes nine steps from the universal triterpenoid progenitor (3S)−2,3-oxidosqualene (Fig. 4). Following the cyclization to the tetracyclic protosta-17(20)Z,24-dien-3β-ol (2) by HelA, HelB1-mediated and HelB2-mediated oxidation at C-4 and C-16, HelD2-dependent acetylation of 16-OH, oxidation of C-21 by HelB4, and HelC-dependent oxidative decarboxylation yield the fusidane skeleton 14, which is further modified in three additional steps mediated by HelB3, HelD1, and HelE to give helvolic acid (1) (Fig. 4). Compared with the late stages in the biosynthesis of helvolic acid, enzymes involved in the early stage modifications act in a relatively strict order (Fig. 4). Considering the fact that an intermediate featured with both C-20 carboxyl and C-16β hydroxyl could be readily converted to the off-pathway lactone

derivative[13], it is reasonable that the hydroxylation of C-16 by HelB1 and subsequent acetylation by HelD2 should occur before the HelB3-mediated oxidation of C-21.

Different from the common sterol biosynthesis, biosynthesis of helvolic acid involves a distinct C-4 demethylation process (Fig. 5a, b). In the biosynthesis of sterol, it is the methyl group at C-4α position that is oxidized to carboxylic acid by SMO and subsequently eliminated by oxidative decarboxylation under the action of 3βHSD to yield a transient enol intermediate (Fig. 5b), which should be rapidly converted to the 3-keto tautomer. In this process, the remaining methyl group originally at 4β (axial) position will energetically favor epimerization to 4α (equatorial) position, which provides a second chance for oxidation by SMO and undergoes demethylation (Fig. 5b)[25–27]. Nevertheless, C-4 demethylation in fusidane-type antibiotics proceeds in an unusual manner though it is also achieved by oxidative decarboxylation. Feeding experiments using[14]C- and [3]H-labeled mevalonic acid revealed that formation of fusidic acid involves the direct loss of the C-4β methyl group[9]. Reconstitution of the early stage biosynthesis of helvolic acid in *S. cerevisiae* revealed that HelB1 specifically converts C-4β methyl group into carboxyl group, and HelC catalyzes the formation of a 3-keto derivative[18]. In the present study, our experiments demonstrated that it is the methyl group at C-4β position that is oxidized by HelB1 and subsequently removed by a promiscuous SDR enzyme HelC (Fig. 5a). During C-4β demethylation, the remaining C-4α methyl group originally at the equatorial position is energetically favorable and difficult to epimerize to the β position for further oxidation by HelB1 (Fig. 5a). This could be the reason why fusidane-type antibiotics retain a single methyl group at their C-4 positions.

In addition, compared with the sterol biosynthesis, in which elimination of C-4α methyl groups is achieved by the successive action of SMO and 3βHSD (Fig. 5b), in the biosynthesis of helvolic acid, after HelB1-mediated oxidation of C-4β methyl, HelC-catalyzed decarboxylation can take place at different biosynthetic stages as a result of the promiscuity of HelC (Fig. 4). Interestingly, premature decarboxylation can prevent the substrates from being modified by HelB2, HelD2, and HelB4, implying that the C-4β carboxylic acid or the 3β-hydroxyl might be important for the recognition by these enzymes in the early stage. In contrast, HelE and HelB3 can only work on the decarboxylated products (Fig. 4). Considering that the reaction sites of these two enzymes are on the A-ring or near the A-ring, the axial C-4β carboxylic acid may sterically block the recognition by these enzymes.

There are four P450 enzymes in the biosynthetic gene cluster for helvolic acid. Among them, HelB3 is able to catalyze the dual oxidation of C-6 and C-7, to which a similar mechanism is only observed in the fungal meroterpenoid terretonin biosynthesis[28]. Sequence alignment of the P450 enzymes from the helvolic acid gene cluster reported by us and those identified from *Sarocladium oryzae*[29] and *Metarhizium anisopliae*[30] revealed some amino acid residues are specifically conserved in HelB3 homologs, but not in other P450 enzymes (Supplementary Fig. 27), which might account for its intriguing property. Additional X-ray crystal structure analysis is needed to provide deep insight into the enzymatic reaction mechanisms.

In this study, we obtained helvolic acid (~20 mg L$^{-1}$) and its 21 derivatives via reconstitution in *A. oryzae* NSAR1. We investigated the antibacterial activity of all these compounds and found that the C-20 carboxyl group and 3-keto are essential for their antimicrobial activity against *S. aureus*, which is consistent with the previous report[13]. It is worth noting that three intermediates 17, 19 and 22 exhibit more potent antibacterial activity than the end product helvolic acid. Based on the structural difference between 17 (MIC: 1 μg mL$^{-1}$) and 20 (MIC: 16 μg mL$^{-1}$), and 22

### Table 2 Anti-*Staphylococcus aureus* activity of compounds

| Compound | MIC (μg mL$^{-1}$) | Compound | MIC (μg mL$^{-1}$) |
|---|---|---|---|
| Helvolic acid (1) | 2 | 12 | >128 |
| 2 | >128 | 13 | >128 |
| 3 | 128 | 14 | 2 |
| 4 | >128 | 15 | 8 |
| 5 | >128 | 16 | 64 |
| 6 | 32 | 17 | 1 |
| 7 | >128 | 18 | 32 |
| 8 | 64 | 19 | 1 |
| 9 | >128 | 20 | 16 |
| 10 | 16 | 21 | 32 |
| 11 | 32 | 22 | 0.5 |

(MIC: 0.5 µg mL$^{-1}$) and helvolic acid (MIC: 2 µg mL$^{-1}$), we could conclude that the free C-6 hydroxyl group is important for the antibacterial activity and further acetylation of 6β-OH is deleterious to the activity. These results also give a good example that in some cases, not all the biosynthetic genes in the cluster are designed to increase the biological activities of secondary metabolites, which is also shown in the cylindrocyclophane biosynthesis[31]. Our results suggested that the biosynthetic intermediates and shunt products might be an invaluable source of drug discovery. However, natural products obtained via the conventional method are usually the end products biosynthesized in the hosts[32], and access to the biosynthetic intermediates is always difficult. Therefore, biosynthetic approaches, especially heterologous expression, could allow the isolation of as many intermediates as possible[23, 33], which will be an effective way to obtain more active derivatives.

Increased emergence of bacterial resistance has evoked the attention to fusidane-type antibiotics[34], and the unambiguous determination of function and reaction sequences of nine genes in the helvolic acid gene cluster will advance the generation of artificial fusidane analogs using synthetic biology. The distinct C-4 demethylation mechanism identified in the helvolic acid biosynthesis could be used for generating single C-4 methylated triterpenoids. Moreover, the stepwise reconstitution of the helvolic acid biosynthesis in A. oryzae has enabled us to isolate twenty-two products and obtain three intermediates with more potent activity than helvolic acid. We believe that heterologous expression could not only be useful for exploration of biosynthetic pathway, but also a powerful approach to complement conventional isolation in terms of drug discovery.

In conclusion, we have systematically unraveled the complete biosynthetic pathway to helvolic acid, one of the most representative fusidane-type antibiotics with potent activity against Gram-positive bacteria, and elucidated an unusual C-4 demethylation process that is distinct from the common sterol biosynthesis. In addition, we have obtained, via reconstitution in A. oryzae NSAR1, helvolic acid and its 21 derivatives, three of which are more active than helvolic acid. These findings provide knowledge of the biosynthesis of fusidane-type antibiotics and demonstrate the applicability of synthetic biology to generate analogs of helvolic acid.

## Methods

**General materials and experimental procedures.** Acetonitrile (CH$_3$CN) was purchased from Oceanpak Alexative Chemical Co., Ltd. (Gothenburg, Sweden). Petroleum ether, ethyl acetate (EtOAc), and acetone were analytical-grade from Fine Chemical Co., Ltd. (Tianjin, China). Formic acid was purchased from Kemiou Chemical Reagent Co., Ltd. (Tianjin, China).

Primer synthesis and DNA sequencing were performed by Sangon Biotech Co., Ltd. (Shanghai, China). Plasmid purification kits and agarose gel DNA extraction kits were purchased from Sangon Biotech Co., Ltd. (Shanghai, China). PCR was carried out using a Mastercycler nexus gradient (Eppendorf, Hamburg, Germany) with KOD -Plus- polymerase (Toyobo, Osaka, Japan). In-Fusion® HD Cloning Kit and T4 DNA polymerase were purchased from Takara Bio Inc. (Dalian, China). Other DNA modification reagents were purchased from Thermo Fisher Scientific Inc. (Shenzhen, China).

GC–MS analysis was performed on Thermo Finnigan Trace GC Ultra equipped with a HP-5MS column (0.25 mm i.d. × 30 m, 0.25 µm film thickness) and coupled with Thermo Finnigan Trace DSQ. The temperature of the ionization chamber was 300 °C and the electron impact ionization voltage was 70 eV. The oven temperature began at 75 °C and held for 5 min, then increased to 230 °C at a rate of 20 °C min$^{-1}$ and stayed at 230 °C for 2 min. It then continued with a second ramp by rising at a rate of 10 °C min$^{-1}$ to 310 °C and held for 10 min. Helium was used as the carrier gas.

HPLC and LC–MS were carried out on an Ultimate 3000 HPLC system (Dionex) and an amaZon SL ion trap mass spectrometer (Bruker) using electrospray ionization with a Cosmosil 5C$_{18}$-MS-II column (4.6 mm i.d. × 250 mm, 5 µm; Nacalai Tesque, Inc., Japan). Elution was subjected to a linear gradient [H$_2$O containing 0.1% formic acid (A) and CH$_3$CN containing 0.1% formic acid (B); 1 mL min$^{-1}$; 50%-100% B (0-30 min), 100% B (30–70 min); 208 nm].

UV data, IR data and optical rotations were respectively measured on the JASCO V-550 UV/vis spectrometer, JASCO FT/IR-480 plus spectrometer, and JASCO P1020 digital polarimeter from JASCO International Co., Ltd., Tokyo, Japan. The HRESIMS data were obtained on a Micromass Q-TOF mass spectrometer (Waters Corporation, Milford, USA). 1D and 2D NMR spectra were obtained with Bruker AV 400 and Bruker AV 600 spectrometers (Bruker BioSpin Group, Faellanden, Switzerland) using the solvent signals (CDCl$_3$: $\delta_H$ 7.26/$\delta_C$ 77.0; Pyridine-$d_5$: $\delta_H$ 7.21/$\delta_C$ 123.5) as internal standards.

The semi-preparative HPLC was performed on an Ultimate 3000 HPLC system (Dionex) using a YMC-Pack ODS-A column (10.0 mm i.d. × 250 mm, 5 µm; YMC Co., Ltd., Kyoto, Japan). Medium-pressure liquid chromatography (MPLC) was equipped with a dual-pump gradient system, a UV preparative detector, and a Dr. Flash II fraction collector system (Lisui E-Tech Co., Ltd., Shanghai, China). Column chromatography (CC) was performed with silica gel (200–300 mesh, Haiyang Chemical Co., Ltd., Qingdao, China) and ODS (50 µm, YMC Co., Ltd., Tokyo, Japan).

**Strains and media.** Aspergillus oryzae NSAR1 (niaD$^-$, sC$^-$, ΔargB, adeA$^-$)[20] served as the host for heterologous expression of genes. Transformants of A. oryzae NSAR1 were grown in 10 mL DPY medium (2% dextrin, 1% polypeptone, 0.5% yeast extract, 0.05% MgSO$_4$·7H$_2$O, 0.5% KH$_2$PO$_4$) for 1–2 days at 28 °C and 150 rpm. The cells were then transferred into Czapek-Dox (CD) medium (0.3% NaNO$_3$, 0.2% KCl, 0.05% MgSO$_4$·7H$_2$O, 0.1% KH$_2$PO$_4$, 0.002% FeSO$_4$·7H$_2$O, 1% polypeptone, 2% starch, pH 5.5) to induce the expression of heterologous genes under the amyB promoter.

Standard gene engineering experiments were performed using E. coli DH5α in LB medium supplemented with appropriate antibiotics, and E. coli BL21(DE3) was used for expression of HelC.

**Construction of fungal expression plasmids.** All the nine genes in the helvolic acid gene cluster were amplified from Aspergillus fumigatus Af293 genomic DNA with the primers listed in Supplementary Table 18. The helA gene was ligated into the SmaI-linearized pTAex3 vector using the In-Fusion® HD Cloning Kit to yield pTAex3-helA. The helB1 was cloned into the KpnI site of the pUSA vector to yield pUSA-helB1. The other seven genes were ligated into the pTAex3 vector digested with EcoRI or (and) KpnI to create pTAex3-helB2, pTAex3-helB3, pTAex3-helB4, pTAex3-helC, pTAex3-helD1, pTAex3-helD2, pTAex3-helE. pUSA-helC was constructed by ligation of the fragment amplified from pTAex3-helC using HelC-EcoRI-F/HelC-EcoRI-R to the SmaI-digested pUSA vector.

pUSA-helB1-helC was constructed by ligation of the helC-containing fragment generated from the BamHI-digestion of pTAex3-helC to the BamHI-linearized pUSA-helB1. pAdeA-helB2, pAdeA-helB3, pAdeA-helB4, pAdeA-helE, pPTRI-helB3, and pPTRI-helE were constructed by the insertion of the fragments amplified from pTAex3-helB2, pTAex3-helB3, pTAex3-helB4, pTAex3-helE with the primers pTAex3-Pamy-F/pTAex3-Tamy-R into the XbaI-digested pAdeA or SmaI-digested pPTRI. For construction of plasmids harboring two or three genes, DNA fragments including the amyB promoter and terminator were amplified from pTAex3-based plasmids with the corresponding primers, and then ligated with the XbaI-digested pAdeA or HindIII-digested pPTRI using the In-Fusion® HD Cloning Kit. All of the expression plasmids are listed in Supplementary Table 19.

**Transformation of A. oryzae NSAR1.** The spore suspension of A. oryzae NSAR1 was inoculated into the 10 mL DPY medium and cultivated for two days at 28 °C and 150 rpm. Then the cells were transferred into 100 mL DPY medium and continued growing for one day at 28 °C and 150 rpm. Mycelia were collected by filtration, and cell walls were digested using 1% Yatalase (Takara) in 0.6 M (NH$_4$)$_2$SO$_4$, 50 mM maleic acid, pH 5.5 at 30 °C for 3 h. After removing residues by filtration, protoplasts were centrifuged at 1500 rpm for 10 min and washed with Solution 2 (1.2 M sorbitol, 50 mM CaCl$_2$·2H$_2$O, 35 mM NaCl, 10 mM Tris-HCl, pH 7.5), and then adjusted to 2 × 10$^7$ cells mL$^{-1}$ in Solution 2. Mixtures of 200 µL protoplasts solution and <15 µL plasmids (10 µg) were incubated on ice for 30 min, and subsequently 1.35 mL Solution 3 (60% PEG4000, 50 mM CaCl$_2$·2H$_2$O, 10 mM Tris-HCl, pH 7.5) was added to the aliquot. After 20 min incubation at the room temperature, 10 mL Solution 2 was added, and then the mixture was subjected to centrifugation at 1500 rpm for 10 min. The precipitates were suspended in 200 µL Solution 2 and spread on the lower selective medium (0.2% NH$_4$Cl, 0.1% (NH$_4$)$_2$SO$_4$, 0.05% KCl, 0.05% NaCl, 0.1% KH$_2$PO$_4$, 0.05% MgSO$_4$·7H$_2$O, 0.002% FeSO$_4$·7H$_2$O, 2% glucose and 1.2 M sorbitol as well as 0.15% methionine, 0.1% arginine, 0.01% adenine, and 0.1 µg mL$^{-1}$ pyrithiamine hydrobromide if necessary, pH 5.5) with 1.5% agar, and then covered with the selective upper medium containing 0.8% agar. The plates were incubated at 30 °C for five to seven days. All of the transformants used in the work are listed in Supplementary Table 20.

**Analysis and isolation of metabolites.** To analyze the metabolites in different transformants, the seed broth cultured in 10 mL DPY medium for 2 days were inoculated into 100 mL CD medium in a 500 mL flask at 28 °C for 5–6 days on a rotary shaker at 150 rpm. Then, mycelia were collected by filtration and extracted with acetone at the room temperature, and if necessary, the culture medium was extracted with EtOAc. The extract concentrated under reduced pressure was

resuspended for analysis. For isolation of metabolites, extract was subjected to the column chromatography and further purification by semi-preparative HPLC (Supplementary Methods).

**Heterologous expression and purification of HelC**. The fragment coding HelC amplified from pTAex3-*helC* using SDR-NdeI-F/SDR-HindIII-R was cloned into the expression vector pET28a at the *Nde*I and *Hind*III sites in frame with an *N*-terminal His$_6$-tag sequence. *E. coli* BL21(DE3) harboring pET28a-*helC* was inoculated into 10 mL LB medium with 50 mg L$^{-1}$ kanamycin and cultured at 37 °C and 200 rpm overnight, and then 1.5 mL seed broth was transferred into the 500 mL flask with 150 mL LB medium and 50 mg L$^{-1}$ kanamycin, and grew at 37 °C and 200 rpm until OD$_{600}$ reached up to 0.5–0.7. The gene expression was induced with 0.2 mM isopropyl β-D-1-thiogalactopyranoside (IPTG) at 18 °C for 12–16 h. Cells were harvested by centrifugation at 4 °C and 5000 × *g* for 10 min. The pellet was washed with 50 mM Tris-HCl (pH 8.0) and resuspended in lysis buffer (50 mM Tris-HCl, 200 mM NaCl, 5 mM imidazole, 5% glycerol, 1% proteinase inhibitor cocktail, 0.1% lysozyme, pH 8.0), followed by sonication on ice for 10 min. After centrifugation at 4 °C and 10,000 × *g* for 10 min, the supernatant was further filtered through a 0.45 μm filter to remove cell debris and other particulate materials. The filtrate was subjected to a Ni Sepharose™ 6 Fast Flow (GE Healthcare Bio-Sciences AB, Uppsala, Sweden) affinity chromatography column equilibrated with binding buffer (50 mM Tris-HCl, 200 mM NaCl, 5 mM imidazole, 5% glycerol, pH 8.0). After elution with wash solution (50 mM Tris-HCl, 200 mM NaCl, 20 mM imidazole, 5% glycerol, pH 8.0) to remove host proteins, the bound HelC was eluted with elution solution (50 mM Tris-HCl, 200 mM NaCl, 300 mM imidazole, 5% glycerol, pH 8.0). The fraction containing HelC was concentrated and buffer-exchanged using the Amicon centrifugal filter Ultracel®-10K (Merck Millipore Ltd., Darmstadt, Germany). The purified HelC was flash frozen in liquid nitrogen and stored at −80 °C for further use. In the course of expression and purification, HelC was confirmed using SDS-PAGE. Protein concentration was quantified according to the Bradford method via Pierce™ BCA Protein Assay Kit (Thermo Fisher Scientific Inc., Rockford, USA) and using bovine serum albumin as the standard protein.

**In vitro assays and feeding experiments**. In vitro enzymatic reaction for HelC was carried out in a reaction mixture (final volume, 200 μL) containing 100 mM Tris-HCl (pH 8.0), 5 mM NAD trihydrate, 2.35–7.00 μM HelC, and 0.5 mM substrate. Following incubation at 30 °C overnight, the mixture was extracted with 200 μL EtOAc. After evaporation of solvents, the residue was resuspended in methanol for HPLC analysis.

The *A. oryzae* NSAR1 transformant strain only containing *helB1*, *helB2*, *helB4*, or *helD2* was cultured in 10 mL DPY medium for 1–2 days, and then was inoculated into 50 mL CD medium for expression of exogenous genes. After cultivation for 1 day, the culture was supplemented with 1.0 mg of substrate dissolved in 20 μL DMSO, and was further incubated for 5 days. Mycelia were harvested and extracted with acetone. After removal of solvents under reduced pressure, samples were dissolved in methanol for LC–MS analysis.

**Antibacterial assay**. In vitro antibacterial activity against Gram-positive strain *Staphylococcus aureus* 209 P was evaluated using the 2-fold dilution assay[35, 36]. The bacterial strain was inoculated on nutrient agar (0.3% beef extract, 0.2% yeast extract, 1% peptone, 0.5% NaCl and 1.5% agar, pH 7.5) at 37 °C for 1–2 days, then cells were collected with normal saline and adjusted to a concentration of 10$^7$–10$^9$ mL$^{-1}$ as seed broth. Compounds were dissolved into DMSO and adjusted to 50 mg L$^{-1}$. A total of 100 μL seed broth was added to 200 mL nutrient solution, and then 200 μL of mixtures was transferred into the first test well of each line in the 96-well while 100 μL was added in the other wells in the same line. 0.5 μL sample solution was added into the first well with the concentration of 128 mg L$^{-1}$, then 100 μL was transferred to the second well until the twelfth well with the final concentration of 0.06 mg L$^{-1}$. Tobramycin was used as the positive control and DMSO as the negative control. The 96-well microtiter plates were placed at 37 °C for 24 h. The MIC was defined based on the minimal concentration in which no bacteria were observed.

**Structural characterization**. Compound 1: A white powder; HRESIMS (positive) *m/z* 591.2932 [M + Na]$^+$ (calcd for C$_{33}$H$_{44}$O$_8$Na, 591.2934), see Supplementary Fig. 28a; NMR spectra, see Supplementary Fig. 28b, c; The NMR data are in good agreement with those of 6β,16β-diacetyloxy-29-norprotosta-1,17(20)Z,24-trien-3,7-dione-21-oic acid (helvolic acid[37]) (Supplementary Note 1).

Compound 2: A colorless oil; NMR spectra, see Supplementary Fig. 29 a, b; The NMR data are in good agreement with those of protosta-17(20)Z,24-dien-3β-ol[19] (Supplementary Note 2).

Compound 3: A colorless oil; HRESIMS (positive) *m/z* 425.3793 [M + H]$^+$ (calcd for C$_{30}$H$_{49}$O, 425.3783), see Supplementary Fig. 30a; NMR spectra, see Supplementary Fig. 30b, c; The NMR data are in good agreement with those of protosta-17(20)Z,24-dien-3-one[18] (Supplementary Note 3).

Compound 4: A white powder; HRESIMS (positive) *m/z* 457.3665 [M + H]$^+$ (calcd for C$_{30}$H$_{49}$O$_3$, 457.3682), see Supplementary Fig. 31a; NMR spectra, see Supplementary Fig. 31b–g; NMR data, see Supplementary Table 1; The NMR data

are in agreement with those of 3β-hydroxy-protosta-17(20)Z,24-dien-29-oic acid [protosta-17(20)Z,24-dien-3β-ol-29-oic acid[18]], but are slightly revised (Supplementary Note 4).

Compound 5: A colorless oil; HRESIMS (positive) *m/z* 411.3600 [M + H]$^+$ (calcd for C$_{29}$H$_{47}$O, 411.3627), see Supplementary Fig. 32a; NMR spectra, see Supplementary Fig. 32b–g; NMR data, see Supplementary Table 2; The structure is as the same as that of 29-norprotosta-17(20)Z,24-dien-3-one [3-oxofusida-17(20)Z,24-diene[38]], the detailed NMR assignments of which are not available prior to the present study (Supplementary Note 5).

Compound 6: A white powder; [α]$_D^{23}$ + 52.0 (*c* 0.5, CHCl$_3$); UV (CH$_3$OH) λ$_{max}$ (log ε) 206 (4.05), 252 (3.33); IR (KBr) ν$_{max}$ 3429, 2947, 2872, 1697, 1621, 1455, 1375, 1027 cm$^{-1}$; HRESIMS (positive) *m/z* 473.3622 [M + H]$^+$ (calcd for C$_{30}$H$_{49}$O$_4$, 473.3631), see supplementary Fig. 33a; NMR spectra, see Supplementary Fig. 33b–i; NMR data, see Supplementary Table 3; 6 is identified as 3β,16β-dihydroxy-protosta-17(20)E,24-dien-29-oic acid (Supplementary Note 6).

Compound 7: A colorless oil; [α]$_D^{23}$ + 39.6 (*c* 0.5, CHCl$_3$); UV (CH$_3$OH) λ$_{max}$ (log ε) 205 (3.67), 248 (3.18); IR (KBr) ν$_{max}$ 3421, 2941, 2872, 1705, 1624, 1375, 1046 cm$^{-1}$; HRESIMS (positive) *m/z* 427.3572 [M + H]$^+$ (calcd for C$_{29}$H$_{47}$O$_2$, 427.3576), see Supplementary Fig. 34a; NMR spectra, see Supplementary Fig. 34b–h; NMR data, see Supplementary Table 4; 7 is identified as 16β-hydroxy-29-norprotosta-17(20)E,24-dien-3-one (Supplementary Note 7).

Compound 8: A yellowish powder; [α]$_D^{23}$ + 53.4 (*c* 0.5, CHCl$_3$); UV (CH$_3$OH) λ$_{max}$ (log ε) 206 (3.85), 253 (2.79); IR (KBr) ν$_{max}$ 3445, 2944, 2875, 1729, 1705, 1449, 1378, 1239, 1027 cm$^{-1}$; HRESIMS (positive) *m/z* 537.3557 [M + Na]$^+$ (calcd for C$_{32}$H$_{50}$O$_5$Na, 537.3556), see Supplementary Fig. 35a; NMR spectra, see Supplementary Fig. 35b–h; NMR data, see Supplementary Table 5; 8 is identified as 16β-acetyloxy-3β-hydroxy-protosta-17(20)E,24-dien-29-oic acid (Supplementary Note 8).

Compound 9: A colorless oil; [α]$_D^{23}$ + 37.4 (*c* 0.5, CHCl$_3$); UV (CH$_3$OH) λ$_{max}$ (log ε) 207 (3.65), 248 (3.05); IR (KBr) ν$_{max}$ 2950, 2869, 1708, 1454, 1373, 1245 cm$^{-1}$; HRESIMS (positive) *m/z* 491.3472 [M + Na]$^+$ (calcd for C$_{31}$H$_{48}$O$_3$Na, 491.3501), see Supplementary Fig. 36a; NMR spectra, see Supplementary Fig. 36b–g; NMR data, see Supplementary Table 6; 9 is identified as 16β-acetyloxy-29-norprotosta-17(20)E,24-dien-3-one (Supplementary Note 9).

Compound 10: A white powder; [α]$_D^{23}$ + 136.0 (*c* 0.5, CH$_3$OH); UV (CH$_3$OH) λ$_{max}$ (log ε) 205 (3.77), 246 (2.97); IR (KBr) ν$_{max}$ 3429, 2965, 2938, 2851, 1688, 1466, 1449, 1375, 1098, 1027 cm$^{-1}$; HRESIMS (positive) *m/z* 473.3615 [M + H]$^+$ (calcd for C$_{30}$H$_{49}$O$_4$, 473.3631), see Supplementary Fig. 37a; NMR spectra, see Supplementary Fig. 37b–g; NMR data, see Supplementary Table 7; 10 is identified as 3β,20-dihydroxy-protosta-16,24-dien-29-oic acid with the absolute configuration of C-20 unsolved (Supplementary Note 10).

Compound 11: A white powder; [α]$_D^{23}$ + 109.2 (*c* 0.5, CH$_3$OH); UV (CH$_3$OH) λ$_{max}$ (log ε) 205 (3.91), 249 (3.05); IR (KBr) ν$_{max}$ 3450, 2965, 2938, 2872, 1697, 1469, 1449, 1375, 1098, 1027 cm$^{-1}$; HRESIMS (positive) *m/z* 473.3611 [M + H]$^+$ (calcd for C$_{30}$H$_{49}$O$_4$, 473.3631), see Supplementary Fig. 38a; NMR spectra, see Supplementary Fig. 38b–g; NMR data, see Supplementary Table 8; 11 is identified as 3β,20-dihydroxy-protosta-16,24-dien-29-oic acid with the absolute configuration of C-20 unsolved (Supplementary Note 11).

Compound 12: A yellowish powder; [α]$_D^{23}$ + 80.0 (*c* 0.5, CHCl$_3$); UV (CH$_3$OH) λ$_{max}$ (log ε) 205 (3.84), 246 (2.78); IR (KBr) ν$_{max}$ 3459, 2962, 2941, 2869, 1702, 1452, 1375, 1106 cm$^{-1}$; HRESIMS (positive) *m/z* 427.3569 [M + H]$^+$ (calcd for C$_{29}$H$_{47}$O$_2$, 427.3576), see Supplementary Fig. 39a; NMR spectra, see Supplementary Fig. 39b–g; NMR data, see Supplementary Table 9; 12 is identified as 20-hydroxy-29-norprotosta-16,24-dien-3-one with the absolute configuration of C-20 unsolved (Supplementary Note 12).

Compound 13: A yellowish powder; [α]$_D^{23}$ + 59.0 (*c* 0.5, CHCl$_3$); UV (CH$_3$OH) λ$_{max}$ (log ε) 206 (3.87), 245 (2.48); IR (KBr) ν$_{max}$ 3442, 2960, 2936, 2872, 1702, 1455, 1381, 1109 cm$^{-1}$; HRESIMS (positive) *m/z* 427.3568 [M + H]$^+$ (calcd for C$_{29}$H$_{47}$O$_2$, 427.3576), see Supplementary Fig. 40a; NMR spectra, see Supplementary Fig. 40b–g; NMR data, see Supplementary Table 10; 13 is identified as 20-hydroxy-29-norprotosta-16,24-dien-3-one with the absolute configuration of C-20 unsolved (Supplementary Note 13).

Compound 14: A yellowish powder; [α]$_D^{23}$ + 43.0 (*c* 0.5, CH$_3$OH); UV (CH$_3$OH) λ$_{max}$ (log ε) 206 (3.54), 225 (3.45); IR (KBr) ν$_{max}$ 2957, 2944, 2872, 1729, 1708, 1455, 1373, 1253 cm$^{-1}$; HRESIMS (positive) *m/z* 521.3244 [M + Na]$^+$ (calcd for C$_{31}$H$_{46}$O$_5$Na, 521.3243), see Supplementary Fig. 41a; NMR spectra, see Supplementary Fig. 41b–h; NMR data, see Supplementary Table 11; 14 is identified as 16β-acetyloxy-29-norprotosta-17(20)Z,24-dien-3-one-21-oic acid (Supplementary Note 14).

Compound 15: A yellowish powder; [α]$_D^{23}$ + 39.6 (*c* 0.5, CH$_3$OH); UV (CH$_3$OH) λ$_{max}$ (log ε) 206 (3.56), 222 (3.51); IR (KBr) ν$_{max}$ 3439, 2971, 2944, 2869, 1729, 1708, 1449, 1375, 1250, 1033 cm$^{-1}$; HRESIMS (positive) *m/z* 523.3408 [M + Na]$^+$ (calcd for C$_{31}$H$_{48}$O$_5$Na, 523.3399), see Supplementary Fig. 42a; NMR spectra, see Supplementary Fig. 42b–g; NMR data, see Supplementary Table 12; 15 is identified as 16β-acetyloxy-3β-hydroxy-29-norprotosta-17(20)Z,24-dien-21-oic acid (Supplementary Note 15).

Compound 16: A yellowish powder; [α]$_D^{23}$ + 35.6 (*c* 0.5, CH$_3$OH); UV (CH$_3$OH) λ$_{max}$ (log ε) 206 (3.82), 226 (3.74); IR (KBr) ν$_{max}$ 3456, 2968, 2953, 2869, 1729, 1466, 1375, 1267, 1024 cm$^{-1}$; HRESIMS (positive) *m/z* 545.3474 [M + H]$^+$ (calcd for C$_{32}$H$_{49}$O$_7$, 545.3478), see Supplementary Fig. 43a; NMR spectra, see Supplementary Fig. 43b–g; NMR data, see Supplementary Table 13; 16 is identified

as 16β-acetyloxy-3β-hydroxy-protosta-17(20)Z,24-dien-21,29-dioic acid (Supplementary Note 16).

Compound **17**: A yellowish powder; HRESIMS (positive) $m/z$ 551.2987 [M + Na]$^+$ (calcd for $C_{31}H_{44}O_7Na$, 551.2985), see Supplementary Fig. 44a; NMR spectra, see Supplementary Fig. 44b–g; NMR data, see Supplementary Table 14; The structure is as the same as that of 16β-acetyloxy-6β-hydroxy-29-norprotosta-17(20)Z,24-dien-3,7-dione-21-oic acid (CAS: 1379525–35–5), the detailed NMR assignments of which are not available prior to the present study (Supplementary Note 17).

Compound **18**: A yellowish powder; $[\alpha]_D^{23}$ -16.2 ($c$ 0.5, $CH_3OH$); UV ($CH_3OH$) $\lambda_{max}$ (log $\varepsilon$) 207 (3.86), 225 (3.75); IR (KBr) $\nu_{max}$ 3432, 2953, 2938, 2869, 1708, 1621, 1387, 1355, 1267, 1027 cm$^{-1}$; HRESIMS (positive) $m/z$ 553.3139 [M + Na]$^+$ (calcd for $C_{31}H_{46}O_7Na$, 553.3141), see Supplementary Fig. 45a; NMR spectra, see Supplementary Fig. 45b–g; NMR data, see Supplementary Table 15; **18** is identified as 16β-acetyloxy-3β,6β-dihydroxy-29-norprotosta-17(20)Z,24-dien-7-one-21-oic acid (Supplementary Note 18).

Compound **19**: A white powder; HRESIMS (positive) $m/z$ 497.3238 [M + H]$^+$ (calcd for $C_{31}H_{45}O_5$, 497.3267), see Supplementary Fig. 46a; NMR spectra, see Supplementary Fig. 46b–g; NMR data, see Supplementary Table 16; The structure is as the same as that of 16β-acetyloxy-29-norprotosta-1,17(20)Z,24-trien-3-one-21-oic acid [3-oxo-16β-acetoxyfusida-1,17(20)(16,21-cis),24-trien-21-oic acid[39]], the detailed NMR assignments of which are not available prior to the present study (Supplementary Note 19).

Compound **20**: A white powder; HRESIMS (positive) $m/z$ 593.3081 [M + Na]$^+$ (calcd for $C_{33}H_{46}O_8Na$, 593.3090), see Supplementary Fig. 47a; NMR spectra, see Supplementary Fig. 47b, c; The NMR data are in good agreement with those of 6β,16β-diacetyloxy-29-norprotosta-17(20)Z,24-dien-3,7-dione-21-oic acid (1,2-dihydrohelvolic acid[40]) (Supplementary Note 20).

Compound **21**: A white powder; $[\alpha]_D^{23}$ −30.8 ($c$ 0.5, $CH_3OH$); UV ($CH_3OH$) $\lambda_{max}$ (log $\varepsilon$) 206 (3.71), 223 (3.60); IR (KBr) $\nu_{max}$ 3450, 2977, 2944, 2872, 1747, 1440, 1375, 1259, 1027 cm$^{-1}$; HRESIMS (positive) $m/z$ 573.3428 [M + H]$^+$ (calcd for $C_{33}H_{49}O_8$, 573.3427), see Supplementary Fig. 48a; NMR spectra, Supplementary Fig. 48b–g; NMR data, see Supplementary Table 17; **21** is identified as 6β,16β-diacetyloxy-3β-hydroxy-29-norprotosta-17(20)Z,24-dien-7-one-21-oic acid (Supplementary Note 21).

Compound **22**: A white powder; HRESIMS (positive) $m/z$ 527.3009 [M + H]$^+$ (calcd for $C_{31}H_{43}O_7$, 527.3009), see Supplementary Fig. 49a; NMR spectra, see Supplementary Fig. 49b, c; The NMR data are in good agreement with those of 16β-acetyloxy-6β-hydroxy-29-norprotosta-1,17(20)Z,24-trien-3,7-dione-21-oic acid (helvolinic acid[37]) (Supplementary Note 22).

**Data availability**. All relevant data are available within the article and its Supplementary Information and from the corresponding authors on request.

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

## Acknowledgements

This work was mainly supported by grants from the National Natural Science Foundation of China (3171101305, 31670036, 81422054, 81373306), the 111 Project of Ministry of Education of the People's Republic of China (B13038), the JST/NSFC Japanese-Chinese Collaborative Research Program, Guangdong Natural Science Funds for Distinguished Young Scholar (S2013050014287), Natural Science Foundation of Guangdong Province (2014A030313389), Guangdong Special Support Program (2016TX03R280), Guangdong Province Universities and Colleges Pearl River Scholar Funded Scheme (Hao Gao, 2014), and K. C. Wong Education Foundation (Hao Gao, 2016), and a Grant-in-Aid for Scientific Research from the Ministry of Education, Culture, Sports, Science and Technology, Japan (JSPS KAKENHI Grant Number JP15H01836 and JP16H06443).

## Author contributions

D.H., H.G., I.A. and X.-S.Y. designed the research. J.-M.L. and D.H. performed the experiments. J.-M.L., D.H., H.G., T.K., T.A., G.-D.C., C.-X.W., I.A. and X.-S.Y. analyzed the data, and wrote the paper.

## Additional information

**Competing interests:** The authors declare no competing financial interests.

