## [Peer Review File · Nature Communications]

REVIEWERS' COMMENTS:

Reviewer #1 (Remarks to the Author):

The manuscript entitled "Biosynthesis of helvolic acid and identification of an unusual C-4 demethylation process distinct from sterol biosynthesis" by Lv et al shows the production of helvolic acid, a fungi-derived triterpene antibiotic, by heterologous expression. The authors introduced the biosynthetic gene cluster consisting of nine genes into *Aspergillus oryzae* NSAR1 to produce helvolic acid at a titer of 20 mg/L, unambiguously confirming the gene cluster for helvolic acid biosynthesis. Additionally, the stepwise introduction into *A. oryzae* NSAR1 of the genes involved in helvolic acid biosynthesis produced 21 pathway intermediates and shunt products, 13 of which were new chemical structures. In conjunction with biochemical assays using recombinant purified HelC and feeding experiments to strains expressing HelB2 and HelB4, the authors were able to determine the order in which the biosynthetic gene products function in – often a particularly challenging task. Lastly three of the compounds isolated showed more potent antibacterial activity against *Staphylococcus aureus* than the helvolic acid.

Of note is the observation that the C4 beta methyl group is oxidized and decarboxylated in helvolic acid biosynthesis (evidenced by the stereochemical assignment of 4), unlike cholesterol biosynthesis where C4 alpha methyl is oxidized and decarboxylated. The authors propose an appealing chemical rationale for this difference in oxidative stereoselectivity. In cholesterol biosynthesis the equatorial beta carbon is oxidized. Once eliminated the 4-monomethyl group equilibrates to the thermodynamically more stable equatorial configuration, enabling it to be reoxidized by the same P450 and subsequently decarboxylated. Thus both of geminal methyls are lost. As HelB1 in helvolic acid oxidizes the axial beta methyl group, after decarboxylation, the remaining methyl group will occupy the equatorial alpha position and will not be reoxidized, generating the much less common 4-monomethyl product.

This is a thorough stepwise reconstitution of the helvolic acid pathway that generates new and potentially useful analogues. This work identified several new compounds with increased potency, and will be of interest to those in the field of fungal PKS and novel antibiotic discovery. The atypical C4 beta methyl oxidation catalyzed by HelB1 provides an interesting counterpoint to the typical C4 alpha oxidation in cholesterol biosynthesis and will be of broad interest to the biosynthesis community.

This manuscript should be accepted after the authors address or correct the following minor points:

1. There are a few minor grammatical errors throughout the manuscript. For example the last line of paragraph 1, Page 3 reads " To date, only a few of triterpenoids such as...." Should be "To date, only a few triterpenoids such as..."
2. The discussion should expand on the antimicrobial assays that were performed, and should explicitly state the values obtained from the MIC of compounds 17, 19 and 22.
3. In scheme 1, the no reaction arrow going from 9 to 14 should be adjusted. The red X should be positioned towards the middle of the arrow so that it is more visible for the reader.

Reviewer #2 (Remarks to the Author):

In the present work, the author studied about the complete biosynthetic pathway for fusidane-type antibiotic helvolic acid with thirteen new compounds and distinct C-4 demethylation process. I found that this manuscript is interesting and well designed. I have no big critical point at results of this manuscript.

Major point:

Point of research

- What's the novel and unique point of this research? Biosynthesis pathway? Structures? Activity?
Please explain with point for reader.

Reviewer #3 (Remarks to the Author):

The manuscript by Lv et al. describes the elucidation of the grid biosynthetic pathway for helvolic acid. The functions of the nine biosynthetic genes, *helA*, *helB1*, *helB2*, *helC*, *helB3*, *helD1*, *helB4*, *helD2*, and *helE*, were clarified by the heterologous expression experiments using the *A. oryzae* NSAR1 strain as a host strain and identification of the shunt products and biosynthetic intermediates. All of these biosynthetic products were isolated and their chemical structures were confirmed by HRESIMS and NMR analysis.

The authors further revealed the substrate promiscuity of *HelC* by in vitro analysis using its recombinant enzyme and C-4 demethylation process by *HelC*, thereby producing the demethylated shunts products in the helvolic acid biosynthesis. Interestingly, among the shunt compounds, three new compounds (17, 19 and 22) were found to exhibit stronger antibacterial activities than helvolic acid. Based on these findings, the authors provided an important insight into the structure-activity-relationship of the helvolic acid related compounds; both C-20 carboxyl group and 3-keto were important for the antibacterial activities.

Overall, this manuscript would be a nice contribution for publication in Nature Communication.

Other minor clarifications/edits are suggested to improve the manuscript as follows:

Page 3, line 1 from the bottom

You need to define oxidosqualen cyclase as OSC.

Page 4, line 4 from the bottom

The abbreviation, HRESIMS, should be defined here.

Page 5, line 1

The abbreviation, SDR, should be defined here.

Page 6, line 7

The abbreviation, AT, should be defined here.

Page 8, line 1 - 8

The author need to show the data from SDS-PAGE analysis of the recombinant *HelC* to confirm the purity employed in the study.

Scheme 1

HelD2 is a P450 enzyme? It must be acetyltransferase (AT). Check the two positions for *HelD2*.

Responsive letter to reviewers

REVIEWERS' COMMENTS:

Reviewer #1 (Remarks to the Author):

The manuscript entitled “Biosynthesis of helvolic acid and identification of an unusual C-4 demethylation process distinct from sterol biosynthesis” by Lv et al shows the production of helvolic acid, a fungi-derived triterpene antibiotic, by heterologous expression. The authors introduced the biosynthetic gene cluster consisting of nine genes into *Aspergillus oryzae* NSAR1 to produce helvolic acid at a titer of 20 mg/L, unambiguously confirming the gene cluster for helvolic acid biosynthesis. Additionally, the stepwise introduction into *A. oryzae* NSAR1 of the genes involved in helvolic acid biosynthesis produced 21 pathway intermediates and shunt products, 13 of which were new chemical structures. In conjunction with biochemical assays using recombinant purified HelC and feeding experiments to strains expressing HelB2 and HelB4, the authors were able to determine the order in which the biosynthetic gene products function in – often a particularly challenging task. Lastly three of the compounds isolated showed more potent antibacterial activity against *Staphylococcus aureus* than the helvolic acid.

Of note is the observation that the C4 beta methyl group is oxidized and decarboxylated in helvolic acid biosynthesis (evidenced by the stereochemical assignment of 4), unlike cholesterol biosynthesis where C4 alpha methyl is oxidized and decarboxylated. The authors propose an appealing chemical rationale for this difference in oxidative stereoselectivity. In cholesterol biosynthesis the equatorial beta carbon is oxidized. Once eliminated the 4-monomethyl group equilibrates to the thermodynamically more stable equatorial configuration, enabling it to be reoxidized by the same P450 and subsequently decarboxylated. Thus both of geminal methyls are lost. As HelB1 in helvolic acid oxidizes the axial beta methyl group, after decarboxylation, the remaining methyl group will occupy the equatorial alpha position and will not be reoxidized, generating the much less common 4-monomethyl product.

This is a thorough stepwise reconstitution of the helvolic acid pathway that generates new and potentially useful analogues. This work identified several new compounds with increased potency, and will be of interest to those in the field of fungal PKS and novel antibiotic discovery. The atypical C4 beta methyl oxidation catalyzed by HelB1 provides an interesting counterpoint to the typical C4 alpha oxidation in cholesterol biosynthesis and will be of broad interest to the biosynthesis community.

This manuscript should be accepted after the authors address or correct the following minor points:

1. There are a few minor grammatical errors throughout the manuscript. For example the

last line of paragraph 1, Page 3 reads “ To date, only a few of triterpenoids such as....” Should be “To date, only a few triterpenoids such as...”

Response: Thanks a lot for your notice. We have changed the sentence to “To date, only a few triterpenoids such as ginsenosides and betulinic acid have been demonstrated.” (page 3, line 7 from the top) according to your suggestion. In addition, we also carefully checked the grammatical errors throughout the manuscript and corrected them.

2. The discussion should expand on the antimicrobial assays that were performed, and should explicitly state the values obtained from the MIC of compounds 17, 19 and 22.

Response: We agree with the reviewer and have added some comments on the antimicrobial assay and stated the values obtained from the MIC of compounds in the discussion (page 11, line 13-22 from the top): “In this study, we obtained helvolic acid (~20 mg L⁻¹) and its twenty-one derivatives including thirteen new compounds via reconstitution in *A. oryzae* NSAR1. We investigated the antibacterial activity of all these compounds and found that the C-20 carboxyl group and 3-keto are essential for their antimicrobial activity against *S. aureus*, which is consistent with the previous report. It is worth noting that three intermediates 17, 19 and 22 exhibit more potent antibacterial activity than the end product helvolic acid. Based on the structural difference between 17 (MIC: 1 µg mL⁻¹) and 20 (MIC: 16 µg mL⁻¹), and 22 (MIC: 0.5 µg mL⁻¹) and helvolic acid (MIC: 2 µg mL⁻¹), we could conclude that the free C-6 hydroxyl group is important for the antibacterial activity and further acetylation of 6β-OH is deleterious to the activity.”

3. In scheme 1, the no reaction arrow going from 9 to 14 should be adjusted. The red X should be positioned towards the middle of the arrow so that it is more visible for the reader.

Response: We appreciate the point by the reviewer #1. We have adjusted the arrow going from 9 to 14 in Figure 4. And the red X has been positioned towards the middle of the arrow. In addition, we have gone through all the figures in the manuscript.

Reviewer #2 (Remarks to the Author):

In the present work, the author studied about the complete biosynthetic pathway for fusidane-type antibiotic helvolic acid with thirteen new compounds and distinct C-4 demethylation process. I found that this manuscript is interesting and well designed. I have no big critical point at results of this manuscript.

Major point:

Point of research, What's the novel and unique point of this research? Biosynthesis pathway? Structures? Activity? Please explain with point for reader.

Response: Thanks a lot for the point by reviewer #2. There are two major contributions made in the present work, we have incorporated these comments into the discussion section (page12, line 11-18 from the top): “**In conclusion, we have systematically unraveled the complete biosynthetic pathway to helvolic acid, one of the most representative fusidane-type antibiotics with potent activity against Gram-positive bacteria, and elucidated an unusual C-4 demethylation process that is distinct from the common sterol biosynthesis. In addition, we have obtained, via reconstitution in *A. oryzae* NSAR1, helvolic acid and its twenty-one derivatives including thirteen new compounds, three of which are more active than helvolic acid. These findings provide new knowledge of the biosynthesis of fusidane-type antibiotics and demonstrate the applicability of synthetic biology to generate analogs of helvolic acid.**”

Reviewer #3 (Remarks to the Author):

The manuscript by Lv et al. describes the elucidation of the grid biosynthetic pathway for helvolic acid. The functions of the nine biosynthetic genes, helA, helB1, helB2, helC, helB3, helD1, helB4, helD2, and helE, were clarified by the heterologous expression experiments using the *A. oryzae* NSAR1 strain as a host strain and identification of the shunt products and biosynthetic intermediates. All of these biosynthetic products were isolated and their chemical structures were confirmed by HRESIMS and NMR analysis.

The authors further revealed the substrate promiscuity of HelC by in vitro analysis using its recombinant enzyme and C-4 demethylation process by HelC, thereby producing the demethylated shunts products in the helvolic acid biosynthesis. Interestingly, among the shunt compounds, three new compounds (17, 19 and 22) were found to exhibit stronger antibacterial activities than helvolic acid. Based on these findings, the authors provided an important insight into the structure-activity-relationship of the helvolic acid related compounds; both C-20 carboxyl group and 3-keto were important for the antibacterial activities.

Overall, this manuscript would be a nice contribution for publication in Nature Communication.

Other minor clarifications/edits are suggested to improve the manuscript as follows:

Page 3, line 1 from the bottom, You need to define oxidosqualen cyclase as OSC.

Response: Thanks for the notice by reviewer #3. We have defined the oxidosqualene cyclase as OSC as the reviewer suggested (Page 4, line 1 from the top). Additionally, we checked the abbreviations throughout the manuscript and defined them.

Page 4, line 4 from the bottom, The abbreviation, HRESIMS, should be defined here.

Response: We have defined high resolution electrospray ionization mass spectrometry as HRESIMS (Page 4, line 4 from the bottom).

Page 5, line 1, The abbreviation, SDR, should be defined here.

Response: We appreciate the point by reviewer #3. We have defined short chain dehydrogenase/reductase as SDR (Page 4, line 3 from the top).

Page 6, line 7, The abbreviation, AT, should be defined here.

Response: We appreciate the point by reviewer #3. We have defined acyltransferase as AT (Page 6, line 7 from the top).

Page 8, line 1 – 8, The author need to show the data from SDS-PAGE analysis of the recombinant HeIC to confirm the purity employed in the study.

Response: We have added the SDS-PAGE analysis of the recombinant HeIC as a supplementary Figure 26 (Page 7, line 1 from the bottom)

Scheme 1 HeID2 is a P450 enzyme? It must be acetyltransferase (AT). Check the two positions for HeID2.

Response: We appreciate the point by reviewer #3. We have corrected HeID2 as acyltransferase (AT) in Figure 4. Additionally, we have gone through all the figures.